# R-Drop: Regularized Dropout for Neural Networks

**Xiaobo Liang**[1]* **Lijun Wu**[2]* **Juntao Li**[1] **Yue Wang**[1] **Qi Meng**[2]
**Tao Qin**[2] **Wei Chen**[2] **Min Zhang**[1] **Tie-Yan Liu**[2]
[1]Soochow University, [2]Microsoft Research Asia
xbliang3@stu.suda.edu.cn, {ljt,minzhang}@suda.edu.cn, wangyuenlp@gmail.com
{lijuwu,meq,taoqin,wche,tyliu}@microsoft.com

## Abstract

Dropout is a powerful and widely used technique to regularize the training of deep neural networks. Though effective and performing well, the randomness introduced by dropout causes unnegligible inconsistency between training and inference. In this paper, we introduce a simple consistency training strategy to regularize dropout, namely R-Drop, which forces the output distributions of different sub models generated by dropout to be consistent with each other. Specifically, for each training sample, R-Drop minimizes the bidirectional KL-divergence between the output distributions of two sub models sampled by dropout. Theoretical analysis reveals that R-Drop reduces the above inconsistency. Experiments on **5** widely used deep learning tasks (**18** datasets in total), including neural machine translation, abstractive summarization, language understanding, language modeling, and image classification, show that R-Drop is universally effective. In particular, it yields substantial improvements when applied to fine-tune large-scale pre-trained models, e.g., ViT, RoBERTa-large, and BART, and achieves state-of-the-art (SOTA) performances with the vanilla Transformer model on WMT14 English→German translation (**30.91** BLEU) and WMT14 English→French translation (**43.95** BLEU), even surpassing models trained with extra large-scale data and expert-designed advanced variants of Transformer models. Our code is available at GitHub[2].

## 1 Introduction

In recent years, deep learning has achieved remarkable success in various areas, e.g., natural language processing, computer vision, speech/audio processing, etc. When training a deep neural network, regularization techniques [57, 60, 27, 3, 67, 58, 23, 71] are indispensable to prevent over-fitting and improve the generalization ability of deep models. Among them, the dropout technique [24], the most widely used one, aims to prevent co-adaptation and performs implicit ensemble by simply dropping a certain proportion of hidden units from the neural network during training. Existing literature [40, 77] has revealed the possible side effect of dropout that there is an unnegligible inconsistency between training and inference stage of dropout models, i.e., the randomly sampled sub model (caused by dropout) during training is inconsistent with the full model (without dropout) during inference. Through imposing $L_2$ regularization on the inconsistent hidden states [40, 77], current methods can mitigate the inconsistency problem to some extent but are far from being widely used.

In this paper, we introduce a simple yet more effective alternative to regularize the training inconsistency induced by dropout, named as R-Drop. Concretely, in each mini-batch training, each data sample goes through the forward pass twice, and each pass is processed by a different sub model by randomly dropping out some hidden units. R-Drop forces the two distributions for the same

---

*Equal contribution (listed in alphabetical order).
[2]https://github.com/dropreg/R-Drop

35th Conference on Neural Information Processing Systems (NeurIPS 2021).

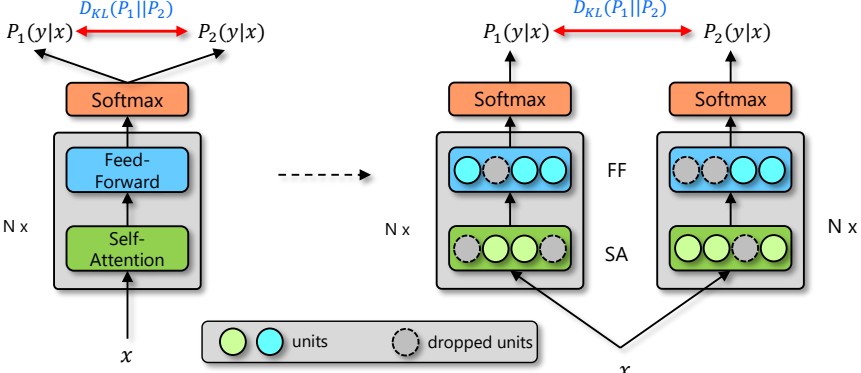

Figure 1: The overall framework of our proposed R-Drop. We take Transformer [59] structure for illustration. The left picture shows that one input $x$ will go through the model twice and obtain two distributions $\mathcal{P}_1$ and $\mathcal{P}_2$, while the right one shows two different sub models produced by dropout.

data sample outputted by the two sub models to be consistent with each other, through minimizing the bidirectional Kullback-Leibler (KL) divergence between the two distributions. That is, R-Drop regularizes the outputs of two sub models randomly sampled from dropout for each data sample in training. In this way, the inconsistency between the training and inference stage can be alleviated. Compared with the dropout strategy in conventional neural network training, R-Drop only adds a KL-divergence loss without any structural modifications.

From the perspective of deep neural network regularization, our proposed R-Drop can be treated as a new variation of dropout. Different from most of the previous methods that merely work on the hidden units of each layer (e.g., the standard dropout [24]) or model parameters (e.g., dropconnect [60]), R-Drop works on both the hidden units and the output of sub models sampled by dropout, which is much more effective. We theoretically analyze the regularization effect of R-Drop, where the result shows that R-Drop can reduce the inconsistency existed in the training and inference.

Though R-Drop regularization is simple, we find it is surprisingly effective through extensive experiments on **5** tasks with **18** datasets, spanning from natural language processing, including language modeling, neural machine translation, abstractive summarization, and language understanding, to computer vision, i.e., image classification. It creates new records on multiple datasets, such as **30.91** BLEU score on WMT14 English→German and **43.95** on WMT14 English→French translation tasks while only be simply applied to the training of the vanilla Transformer, and also achieves SOTA results on the CNN/DailyMail summarization dataset. These universal improvements clearly demonstrate the effectiveness of R-Drop.

Our main contributions are summarized as follows:

- We propose R-Drop, a simple yet effective regularization method built upon dropout, which can be universally applied to train different kinds of deep models.
- We theoretically show that our R-Drop can reduce the inconsistency between training and inference of the dropout based models.
- Through extensive experiments on $4$ NLP and $1$ CV tasks with a total of $18$ datasets, we show that R-Drop achieves extremely strong performances, including multiple SOTA results.

## 2   Approach

The overall framework of our R-Drop regularization method is shown in Figure 1. Before elaborating on the details, we first present some necessary notations. Given the training dataset $\mathcal{D} = \{(x_i, y_i)\}_{i=1}^{n}$, the goal of the training is to learn a model $\mathcal{P}^w(y|x)$, where $n$ is the number of the training samples, $(x_i, y_i)$ is the labeled data pair. $x_i$ is input data and $y_i$ is the label. For example, in NLP, $x_i$ can be the source language sentence in machine translation, and $y_i$ is the corresponding target language sentence. In CV, $x_i$ can be one image, and $y_i$ is the categorical class label. The probability distribution of the

mapping function is also denoted as $\mathcal{P}^w(y|x)$, and the Kullback-Leibler (KL) divergence between two distributions $\mathcal{P}_1$ and $\mathcal{P}_2$ is represented by $\mathcal{D}_{KL}(\mathcal{P}_1||\mathcal{P}_2)$. In the following, we will explain our proposed R-Drop, training algorithm, and theoretical analysis, respectively.

## 2.1 R-Drop Regularization

We introduce our simple regularization method in this part. Given the training data $\mathcal{D} = \{(x_i, y_i)\}_{i=1}^n$, the main learning objective for a deep learning model is to minimize the negative log-likelihood loss function, which is as follow:

$$\mathcal{L}_{nll} = \frac{1}{n} \sum_{i=1}^n -\log \mathcal{P}^w(y_i|x_i). \tag{1}$$

Since the deep neural networks are prone to over-fitting, regularization methods such as dropout [57] are usually adopted during training to reduce the generalization error of the model. Specifically, dropout randomly drops part of units in each layer of the neural network to avoid co-adapting and over-fitting. Besides, dropout also approximately performs to combine exponentially many different neural network architectures efficiently [57], while model combination can always improve the model performance. Though simple and effective, there is a huge inconsistency between training and inference that hinders the model performance. That is, the training stage takes the sub model with randomly dropped units, while the inference phase adopts the full model without dropout. Also, the sub models caused by randomly sampled dropout units are also different without any constraints. Based on above observations and the randomness of the structure brought by dropout, we propose our R-Drop to regularize the output predictions of sub models from dropout.

Concretely, given the input data $x_i$ at each training step, we feed $x_i$ to go through the forward pass of the network twice. Therefore, we can obtain two distributions of the model predictions, denoted as $\mathcal{P}_1^w(y_i|x_i)$ and $\mathcal{P}_2^w(y_i|x_i)$. As discussed above, since the dropout operator randomly drops units in a model, the two forward passes are indeed based on two different sub models (though in the same model). As shown in the right part of Figure 1, the dropped units in each layer of the left path for the output prediction $\mathcal{P}_1^w(y_i|x_i)$ are different from that of the right path for output distribution $\mathcal{P}_2^w(y_i|x_i)$. Thus the distributions of $\mathcal{P}_1^w(y_i|x_i)$ and $\mathcal{P}_2^w(y_i|x_i)$ are different for the same input data pair $(x_i, y_i)$. Then, at this training step, our R-Drop method tries to regularize on the model predictions by minimizing the bidirectional Kullback-Leibler (KL) divergence between these two output distributions for the same sample, which is:

$$\mathcal{L}_{KL}^i = \frac{1}{2}(\mathcal{D}_{KL}(\mathcal{P}_1^w(y_i|x_i)||\mathcal{P}_2^w(y_i|x_i)) + \mathcal{D}_{KL}(\mathcal{P}_2^w(y_i|x_i)||\mathcal{P}_1^w(y_i|x_i))). \tag{2}$$

With the basic negative log-likelihood learning objective $\mathcal{L}_{NLL}^i$ of the two forward passes:

$$\mathcal{L}_{NLL}^i = -\log \mathcal{P}_1^w(y_i|x_i) - \log \mathcal{P}_2^w(y_i|x_i), \tag{3}$$

the final training objective is to minimize $\mathcal{L}^i$ for data $(x_i, y_i)$:

$$\begin{aligned} \mathcal{L}^i = \mathcal{L}_{NLL}^i + \alpha \cdot \mathcal{L}_{KL}^i = &-\log \mathcal{P}_1^w(y_i|x_i) - \log \mathcal{P}_2^w(y_i|x_i) \\ &+ \frac{\alpha}{2}[\mathcal{D}_{KL}(\mathcal{P}_1^w(y_i|x_i)||\mathcal{P}_2^w(y_i|x_i)) + \mathcal{D}_{KL}(\mathcal{P}_2^w(y_i|x_i)||\mathcal{P}_1^w(y_i|x_i))], \end{aligned} \tag{4}$$

where $\alpha$ is the coefficient weight to control $\mathcal{L}_{KL}^i$. In this way, our R-Drop further regularizes the model space beyond dropout and improves the generalization ability of a model. Compared Equation (1) with Equation (4), our R-Drop only adds a KL-divergence loss $\mathcal{L}_{KL}^i$ based on two forward passes in training. Note that our regularization methodology can be universally applied on different model structures if there exists randomness in a model (e.g., dropout) that can produce different sub models or outputs. We leave further explorations as future work.

## 2.2 Training Algorithm

The overall training algorithm based on our R-Drop is presented in Algorithm 1. As introduced before, at each training step, Line 3-4 show that we go forward the model and obtain output distributions $\mathcal{P}_1^w(y|x)$ and $\mathcal{P}_2^w(y|x)$, then Line 5-6 calculate the negative log-likelihood and the KL-divergence

**Algorithm 1** R-Drop Training Algorithm
___
**Input**: Training data $\mathcal{D} = \{(x_i, y_i)\}_{i=1}^n$.
**Output**: model parameter $w$.
1: Initialize model with parameters $w$.
2: **while** not converged **do**
3:     randomly sample data pair $(x_i, y_i) \sim \mathcal{D}$,
4:     repeat input data twice as $[x_i; x_i]$ and obtain the output distribution $[\mathcal{P}_1^w(y_i|x_i), \mathcal{P}_2^w(y_i|x_i)]$,
5:     calculate the negative log-likelihood loss $\mathcal{L}_{NLL}^i$ by Equation (3),
6:     calculate the KL-divergence loss $\mathcal{L}_{KL}^i$ by Equation (2),
7:     update the model parameters by minimizing loss $\mathcal{L}^i$ of Equation (4).
8: **end while**
___

between the two distributions. It is worth nothing that we do not forward the input data twice, instead, we repeat the input data $x$ and concatenate them ($[x; x]$) in batch-size dimension, which can make forward procedure happen in the same mini-batch to save the training cost. Finally, the model parameters are updated (Line 7) according to the loss of Equation (4). The training will continue over the data epochs till convergence. Compared to the conventional training, our implementation is similar to enlarge the batch size to be double, and one potential limitation is that the computational cost of R-Drop increases at each step. As we show in Section 4.1, similar to other regularization methods (e.g., training w/ or w/o dropout), though R-Drop needs more training to converge, the final optimum is much better with a superior performance. We also show another study of baseline with doubled batch size in Appendix C.1.

## 2.3 Theoretical Analysis

We analyze the regularization effect of R-Drop in this subsection. Let $h^l(x) \in \mathbb{R}^d$ denote the output of the $l$-th layer of a neural network with input vector $x$, and let $\xi^l \in \mathbb{R}^d$ denote a random vector, each dimension of which is independently sampled from a Bernoulli distribution $B(p)$:

$$\xi_i^l = \begin{cases} 1, & \textit{with probability p}, \\ 0, & \textit{with probability 1-p}. \end{cases}$$

Then the dropout operation on $h^l(x)$ can be represented by $h_{\xi^l}^l(x) = \frac{1}{p}\xi^l \odot h^l(x)$, where $\odot$ denotes the element-wised product. Hence, the output distribution of the neural network with parameter $w$ after applying dropout is $\mathcal{P}_\xi^w(y|x) := \mathtt{softmax}(\mathtt{linear}(h_{\xi^L}^L(\cdots(h_{\xi^1}^1(x_{\xi^0})))))$, where $\xi = (\xi^L, \cdots, \xi^0)$. The objective for R-Drop enhanced training can be formulated as solving the following constrained optimization problem:

$$\min_w \frac{1}{n} \sum_{i=1}^n \mathbb{E}_\xi[-\log \mathcal{P}_\xi^w(y_i|x_i)], \tag{5}$$

$$s.t. \quad \frac{1}{n} \sum_{i=1}^n \mathbb{E}_{\xi^{(1)},\xi^{(2)}}[\mathcal{D}_{KL}(\mathcal{P}_{\xi^{(1)}}^w(y_i|x_i)||\mathcal{P}_{\xi^{(2)}}^w(y_i|x_i)))] \le \epsilon. \tag{6}$$

More precisely, R-Drop optimizes the constrained optimization problem in Equation (5) and Equation (6) in a stochastic manner, i.e., it samples two random vectors $\xi^{(1)}$ and $\xi^{(2)}$ (corresponding to two dropout instantiations) from Bernoulli distribution and one training instance $(x_i, y_i)$, and updates the parameter $w$ according to the stochastic gradient $\nabla_w \mathcal{L}^i$ from Equation (4).

As we presented, one problem for dropout is the inconsistency between the training and inference models. Specifically, the training objective for dropout is the average loss of the sub models, i.e., $\min_w \frac{1}{n} \sum_{i=1}^n \mathbb{E}_\xi[-\log \mathcal{P}_\xi^w(y_i|x_i)]$, while the full model (denoted as $\tilde{P}^w(y|x)$) is used for inference. Our proposed R-Drop enhanced training reduces this inconsistency by forcing the sub structures to be similar. The following proposition uses a linear model to demonstrate that with the constraint in Equation (6), the inconsistency gap between the average loss of sub structures and the loss of the full model can be bounded (detailed proof can be found in Appendix B).

**Proposition 2.1.** *For a linear model $\mathcal{P}^w(y|x) = \mathtt{softmax}(\mathtt{Norm}(w^T x))$ where $\mathtt{Norm}(\cdot)$ denotes the normalization layer and $x \in \mathbb{R}^d$, with the constraint in Equation (6) in the main paper, we have*

$|\mathcal{L}_{nll}(w) - \mathbb{E}_{\xi}[\mathcal{L}_{nll}(w, \xi)]| \leq c\sqrt{\epsilon}$, *where* $\mathcal{L}_{nll}(w), \mathcal{L}_{nll}(w, \xi)$ *are the empirical loss calculated by the full model and a random sub model respectively,* $c$ *is a constant related to the Liptschtz constant of the softmax operator.*

## 2.4 Discussion

The most related works with our R-Drop are ELD [40] and FD [77], which also study the consistency training with dropout. However, R-Drop has key differences with them. (1) The gap control is from different views. ELD works on directly reducing the gap between the sub model with dropout (train) and the expected full model without dropout (inference), while R-Drop and FD are both working on penalizing the discrepancy between the sub models, the superiority of regularizing the sub models has been proved in FD. (2) The regularization efficiency is different. ELD only back-propagates the gradients through sub model without the full model, which is less efficient than R-Drop that updates both sub models. (3) The regularization effect is different. Both ELD and FD use the $L_2$ distance on hidden states as the regularization loss function. However, this is far away from the main training objective that minimizes the negative log-likelihood over model output distribution. The distance of hidden states is not in the same space as the probability distribution since log-softmax hugely affects the optimization. In comparison, R-Drop utilizes the KL-divergence between the output probability distributions as the consistency regularization, which is in the same space as the training objective. More analysis and experimental comparisons are shown in Appendix C.4.

## 3 Experiments

To evaluate our approach and show its universal impact, we conduct experiments on $5$ different tasks, including $4$ natural language processing (NLP) and $1$ computer vision (CV) tasks, which are neural machine translation (NMT) ($6$ datasets), abstractive summarization ($1$ dataset), language understanding ($8$ datasets), language modeling ($1$ dataset), and image classification ($2$ datasets). For convenience, we utilize 'RD' to represent R-Drop in the tables of experimental results hereinafter. More details of experimental settings for each dataset can be found in Appendix A.

### 3.1 Application to Neural Machine Translation

| Model | En→De | De→En | En→Fr | Fr→En | En→Zh | Zh→En | En→Es | Es→En | Avg |
|---|---|---|---|---|---|---|---|---|---|
| Transformer [59] | 28.57 | 34.64 | 35.9 | 36.1 | 26.3 | 18.4 | 39.0 | 40.6 | 32.44 |
| **Transformer + RD** | **30.72** | **37.25** | **38.0** | **38.9** | **28.1** | **19.5** | **41.8** | **43.2** | **34.68** |

Table 1: BLEU scores on $8$ IWSLT machine translation tasks.

We first evaluate the NMT tasks, which is very important in NLP. To best show the effectiveness of our method, experiments are conducted on both low-resource and rich-resource translation tasks.

**Datasets** The datasets of low-resource scenario are from IWSLT competitions, which include IWSLT14 English↔German (En↔De), English↔Spanish (En↔Es), and IWSLT17 English↔French (En↔Fr), English↔Chinese (En↔Zh) translations. The rich-resource datasets come from the widely acknowledged WMT translation tasks, and we take the WMT14 English→German and English→French tasks. The IWSLT datasets contain about $170k$ training sentence pairs, $7k$ valid pairs, and $7k$ test pairs. The WMT data sizes are $4.5M$, $36M$ for En→De and En→Fr respectively, valid and test data are from the corresponding newstest data.

| Method | En→De | En→Fr |
|---|---|---|
| Transformer [59] | 29.12 | 42.69 |
| MUSE [73] | 29.90 | 43.50 |
| Depth Growing [66] | 30.07 | 43.27 |
| Transformer-Admin [37] | 30.10 | 43.80 |
| Data Diversification [47] | 30.70 | 43.70 |
| BERT-fused NMT [76] | 30.75 | 43.78 |
| **Transformer + RD** | **30.91** | **43.95** |

Table 2: BLEU scores on WMT14 En→De and En→Fr machine translation tasks.

**Model & Training** We take the most popular Transformer [59] network as our model structure. The `transformer_iwslt_de_en` and `transformer_vaswani_wmt_en_de_big` are the configurations for IWSLT and WMT translations respectively. The weight $\alpha$ is set as $5$ for all translation tasks. Implementation is developed on Fairseq [48].

**Results** We calculate the BLEU scores on these tasks for evaluation, following [76]. The IWSLT performances are shown in Table 1 and the rich-resource WMT results are in Table 2. First, we can see that our R-Drop achieves more than 2.0 BLEU score improvements on 8 IWSLT translation tasks, which clearly shows the effectiveness of our method. The results on WMT translations are more impressive. After applying our simple method on the basic Transformer network, we achieve the state-of-the-art (**SOTA**) BLEU score on WMT14 En→De (**30.91**) and En→Fr (**43.95**) translation tasks, which surpass current SOTA models, such as the BERT-fused NMT [76] model that leverages large-scale monolingual data, and the Data Diversification [47] method trained with many translation models. Note that R-Drop is complementary to the above methods, and we believe stronger results can be achieved if we apply R-Drop on their methods and better backbone models beyond Transformer.

## 3.2 Application to Language Understanding

**Dataset** We further evaluate our proposed approach on the language understanding tasks by fine-tuning the pre-trained models[3], which are the standard development sets of GLUE [61] benchmark. The GLUE benchmark includes 8 different text classification or regression tasks, which are MNLI, MRPC, QNLI, QQP, RTE, SST-2, STS-B (regression), CoLA. The detailed statistics are in Appendix.

**Model & Training** We take the BERT-base [9] and strong RoBERTa-large [38] pre-trained models as our backbones to perform fine-tuning, which are publicly available. For each task, different random seeds and parameter settings are required, thus we dynamically adjust the coefficient $\alpha$ among $\{0.1, 0.5, 1.0\}$ for each setting. Other configurations are following the previous works [9, 38]. For the regression task STS-B, we use MSE instead of KL-divergence to regularize the outputs (see Appendix for MSE regularization details).

**Results** The evaluation metrics for above 8 tasks are as follows: The result for STS-B is the Pearson correlation; Matthew's correlation is used for CoLA; Other tasks are measured by Accuracy. The results are presented in Table 3. We can see that R-Drop achieves 1.21 points and 0.80 points (on average) improvement over the two baselines BERT-base and RoBERTa-large, respectively, which clearly demonstrate the effectiveness of R-Drop. Specifically, our RoBERTa-large + RD also surpasses the other two strong models: XLNet-large [68] and ELECTRA-large [7], which are specially designed with different model architecture and pre-training task.

| Model | MNLI | MRPC | QNLI | QQP | RTE | SST-2 | STS-B | CoLA | Avg |
|---|---|---|---|---|---|---|---|---|---|
| BERT-base [9] | 83.8 | 85.3 | 90.8 | 91.0 | 68.2 | 92.4 | 89.3 | 62.3 | 82.85 |
| **BERT-base + RD** | 85.5 | 87.3 | 92.0 | 91.4 | 71.1 | 93.0 | 89.6 | 62.6 | **84.06** |
| RoBERTa-large [38] | 90.2 | 90.9 | 94.7 | 92.2 | 86.6 | 96.4 | 92.4 | 68.0 | 88.93 |
| XLNet-large [68] | 90.8 | 90.8 | 94.9 | 92.3 | 85.9 | 97.0 | 92.5 | 69.0 | 89.15 |
| ELECRTA-large [7] | 90.9 | 90.8 | 95.0 | 92.4 | 88.0 | 96.9 | 92.6 | 69.1 | 89.46 |
| **RoBERTa-large + RD** | 90.9 | 91.4 | 95.2 | 92.5 | 88.4 | 96.9 | 92.5 | 70.0 | **89.73** |

Table 3: Fine-tuned model performances on GLUE language understanding benchmark.

## 3.3 Application to Summarization

**Dataset** Abstractive summarization task is to summarize the long sentence/document into a short sequence/sentence (through generation) with the main content remained. For this generation task, we use the CNN/Daily Mail dataset originally introduced by Hermann et al. [22] to evaluate our method. This dataset contains news documents (source), and their corresponding highlights (target) crawled from CNN and Daily Mail website. It contains 287,226 documents for training, 13,368 documents for validation and 11,490 documents for test. We follow [34] to preprocess the dataset.

**Model & Training** To mostly show the effectiveness, we take the super strong pre-trained sequence-to-sequence BART [34] model as our backbone and fine-tune it using our method. In this task, the coefficient weight $\alpha$ is set as 0.7 to control the KL-divergence. For other hyper-parameters, we follow the setting of the original paper [34] without modification.

---

[3]We apply our R-Drop on the fine-tuning stage only in this work. R-Drop can also be applied during pre-training. Due to the computational cost, we leave this as future work.

**Results** The performance is evaluated by ROUGE F1 score [36]. Specifically, we report the unigram ROUGE-1 (RG-1) and bigram ROUGE-2 (RG-2) overlap to assess the informativeness, and the longest common subsequence ROUGE-L (RG-L) score to assess the fluency. The results are shown in Table 4. We can see that R-Drop based training outperforms the fine-tuned BART model by 0.3 points on RG-1 and RG-2 score and achieves the **SOTA** performance. Specifically, our result also surpasses the PEGASUS method [70], which brings a novel self-supervised paradigm carefully designed for summarization, and the previous best work BART+R3F [1], which introduces a parametric noise

| Method | RG-1 | RG-2 | RG-L |
|---|---|---|---|
| Transformer [59] | 39.50 | 16.06 | 36.63 |
| ProphetNet [51] | 44.02 | 21.17 | **41.30** |
| BART [34] | 44.16 | 21.28 | 40.90 |
| PEGASUS [70] | 44.17 | 21.47 | 41.11 |
| BART + R3F [1] | 44.38 | 21.53 | 41.17 |
| **BART + RD** | **44.51** | **21.58** | 41.24 |

Table 4: ROUGE results on CNN/Daily Mail summarization dataset. RG-1, RG-2, RG-L stand for ROUGE-1, ROUGE-2, and ROUGE-L scores.

sampled from normal or uniform distributions. Instead, our R-Drop does not introduce any extra parameters or model structure changes during training.

## 3.4 Application to Language Modeling

**Dataset** We also evaluate our approach on another widely acknowledged NLP task: language modeling. The dataset we choose for this task is the commonly adopted Wikitext-103 dataset [41], which is the largest available word-level language modeling benchmark with long-term dependency. WikiText-103 contains about $103M$ training tokens from $28K$ articles on Wikipedia, and the average length of tokens per article is about $3.6K$. The data is preprocessed by following [48].

**Model & Training** We take two models to conduct the language modeling task. One is the basic Transformer decoder [59], another is the more advanced one: Adaptive Input Transformer [5], which introduces adaptive input embeddings into the Transformer model. We use the open-source Fairseq [48] toolkit, and the corresponding model configurations are `transformer_lm_gpt` and `transformer_lm_wiki103` for Transformer and Adaptive Input Transformer. We simply set the weight $\alpha$ to be $1.0$ without tuning during training. Other configurations are same as [48] and [5].

**Results** The evaluation metric for language modeling is perplexity, which can well measure the probability of a sentence. Same as [5], we report the perplexity on both valid and test sets. The results are shown in Table 5. From the table, we can see that our R-Drop based training improves the perplexity on both two different model structures, e.g., $0.80$ perplexity improvement on test set over Adaptive Input Transformer. Besides, more improvement can be achieved when the baseline model is not so strong, e.g., $1.79$ perplexity gain on valid set and $1.68$ on test set above the Transformer baseline.

| Method | Valid | Test |
|---|---|---|
| Transformer [59] | 25.76 | 26.62 |
| **Transformer + RD** | **23.97** | **24.94** |
| Adaptive [5] | 18.94 | 18.87 |
| **Adaptive + RD** | **18.18** | **18.07** |

Table 5: Perplexity results on Wikitext-103 language modeling task. Adaptive refers to Adaptive Input Transformer [5].

## 3.5 Application to Image Classification

**Dataset** For image classification, we conduct experiments on two widely acknowledged benchmark datasets, i.e., CIFAR-100 [31] and the ILSVRC-2012 ImageNet dataset [8] (denoted as ImageNet for short). CIFAR-100 dataset consists of $60k$ images of 100 classes, and there are 600 images per class with 500 for training and 100 for testing.

The ImageNet dataset consists of $1.3M$ image samples of $1,000$ categorical classes. We utilize the same data preprocessing strategies with [11], where the details are given in [29].

**Model & Training** We choose the recent strong and popular Vision Transformer (ViT) [11] model as our backbone. More specifically, we take the two publicly released pre-trained models, ViT-B/16 and ViT-L/16, with $86M$ and $307M$ parameters respectively, and we

| Method | CIFAR-100 | ImageNet |
|---|---|---|
| ViT-B/16 [11] | 92.64 | 83.97 |
| **ViT-B/16 + RD** | **93.29** | **84.38** |
| ViT-L/16 [11] | 93.44 | 85.15 |
| **ViT-L/16 + RD** | **93.85** | **85.57** |

Table 6: Accuracy on CIFAR-100 and ImageNet classification tasks.

conduct model fine-tuning on the CIFAR-100 and ImageNet datasets. During fine-tuning, the weight $\alpha$ is set as $0.6$ for both models, and we set other hyper-parameters/training details to be same as [11].

**Results** The classification performance is measured by Accuracy, and the results are presented in Table 6. For CIFAR-100, we achieve about $0.65$ accuracy improvement over ViT-B/16 baseline, and $0.41$ points over ViT-L/16 model. Similarly, on the large-scale ImageNet dataset, consistent improvements are also obtained. These observations demonstrate that our R-Drop can still benefit the model performance even the baseline is powerful. In a word, through the above NLP tasks and this image classification task, we clearly show R-Drop is effective and can be universally applied.

# 4 Study

Beyond the superior experimental results, in this section, we conduct extensive studies on different perspectives to better understand our R-Drop method. The analysis experiments are performed on the IWSLT14 De→En translation task. More studies can be found in Appendix C.

## 4.1 Regularization and Cost Analysis

We first show the regularization effect of our R-Drop and study the potential limitation of training cost (as discussed in Section 2.2). Hence, we plot the curves of training/valid loss and valid BLEU along the training update number for Transformer and Transformer + RD models. Besides, we also plot the corresponding curves along the training time (minutes). The curves are shown in Figure 2. We can observe: 1) Along with the training, Transformer quickly becomes over-fitting, and the gap between train and valid loss of Transformer is large, while R-Drop has a lower valid loss. This well proves that R-Drop can provide persistent regularization during training. 2) At the early training stage, Transformer improves the BLEU score quickly but converges to bad local optima soon. In comparison, R-Drop gradually improves the BLEU score and achieves a much superior performance. Though it needs more training to converge, the final optimum is better. This is same as other regularization methods (e.g., training w/ or w/o dropout). R-Drop indeed increases the training cost at each step since it requires repeating input $x$ for another computation in a mini-batch. Note that this is similar to batch size doubled training without KL-divergence. In Appendix C.1, we conduct this training and show that R-Drop increases negligible cost but with a much stronger performance.

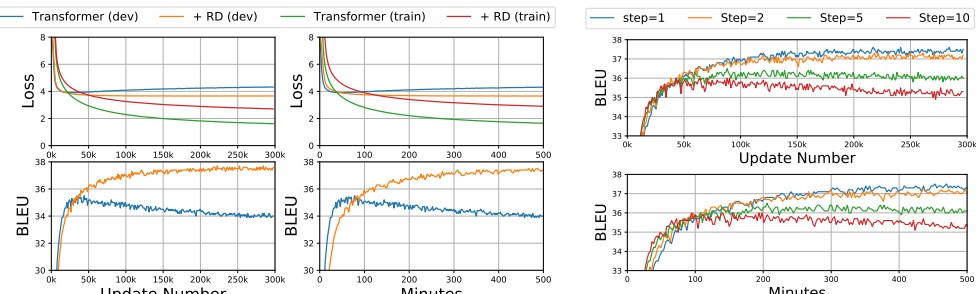

Figure 2: Loss/BLEU curves along with model training.     Figure 3: R-Drop with different step.

## 4.2 $k$-step R-Drop

The above study shows that R-Drop can achieve much stronger performance, but with a lower convergence, thus we study another training strategy that is to perform R-Drop every $k$ steps to improve the training efficiency, instead of applying at each step. We vary $k$ in $\{1, 2, 5, 10\}$ to see the difference, where $k = 1$ is the current training strategy. The valid BLEU curves along with training update number and training time are presented in Figure 3. From the curves, we can conclude that though the convergence is faster with larger $k$, the training fails to fall into good optima, which quickly over-fits, and the BLEU scores become worse and worse when we increase $k$. This proves that our R-Drop at each step can well regularize the training and obtain superior performances.

### 4.3 $m$-time R-Drop

Our method regularizes the model output between two distributions $P_1^w(y|x)$ and $P_2^w(y|x)$, and it is also interesting to see whether more improvements can be achieved if we regularize $m$ distributions for the same input data, where $m = 2$ is the current setting. Therefore, we extend our R-Drop to be: $\mathcal{L}_{KL} = \frac{\alpha}{m*(m-1)} \sum_{i,j \in 1, \cdots, m}^{i \neq j} \mathcal{D}_{KL}(\mathcal{P}_i^w(y|x)||\mathcal{P}_j^w(y|x))$, and we take $m = 3$ for a feasible implementation. The BLEU score for IWSLT14 De→En test set is 37.30 when $m = 3$, which is similar to that when $m = 2$ (37.25 BLEU score). This reflects that R-Drop already has a strong regularization effect between two distributions, without the necessity of stronger regularization.

### 4.4 Two Dropout Rates

Besides the above studies, we investigate R-Drop from another perspective, i.e., the dropout values. In current training, the two distributions are based on the same dropout value (e.g., 0.3 for IWSLT translations). In this study, we utilize two different dropout values for the two output distributions during training (e.g., 0.1 for $P_1^w(y|x)$, 0.3 for $P_2^w(y|x)$) to see the difference. We choose the two dropout rates from $\{0.1, 0.2, 0.3, 0.4, 0.5\}$ with total 15 ($C_5^2$ for two different rates + $C_5^1$ for two same rates) combinations. The results are shown in Figure 4. Among these different results, we can see that: 1) Dropout rates with the same value $(0.3, 0.3)$ is the best choice (current setting), 2) R-Drop

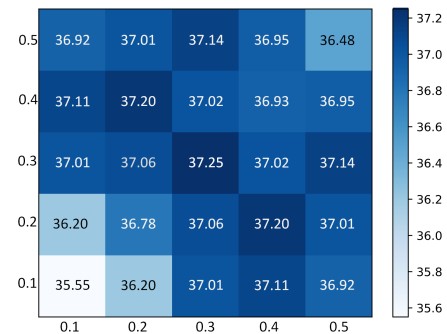

Figure 4: R-Drop with two different dropout rate combinations. Among the 25 numbers, 15 are different since the table is symmetric and triangular.

can stably achieve strong results when the two dropout rates are in a reasonable range $(0.3 \sim 0.5)$ without a big performance difference. One interesting point is that even the two dropout values are both 0.5, which means half of the units are expected to be dropped, R-Drop can still obtain a satisfied result (36.48 BLEU) compared with the baseline Transformer (34.64 BLEU). These results all confirm the advantage of our R-Drop, and we are interested in studying more in the future.

### 4.5 Effect of Weight $\alpha$

Further, we investigate the impact of the KL-divergence loss weight $\alpha$. As mentioned in Section 3.1, we set $\alpha = 5$ for NMT experiments. Here we vary the $\alpha$ in $\{1, 3, 5, 7, 10\}$ and conduct experiments. As shown in Table 7, small $\alpha$ (e.g., 1) can not perform as good as large $\alpha$ (e.g., 5), which means we should pay more attention to the KL-divergence regularization. However, too much regularization ($\alpha = 10$) is also not good, and the best balanced choice is $\alpha = 5$. Note that the choice of $\alpha$ is distinct for different tasks (e.g., NMT, language understanding), which depends on how easy the over-fitting happens caused by the specific data size and model size of each task.

| $\alpha$ | TF + RD |
|---|---|
| $\alpha = 1$ | 36.05 |
| $\alpha = 3$ | 36.85 |
| $\alpha = 5$ | **37.25** |
| $\alpha = 7$ | 37.20 |
| $\alpha = 10$ | 36.95 |

Table 7: BLEU scores with different $\alpha$.

## 5 Related Work

**Regularization Methods.** Bigger models always tend to have better performance, especially for various large-scale pre-trained models, e.g., Vision Transformer [11], Swin Transformer [39], GPT families [52, 53, 6], BERT [9], BART [34], Switch Transformers [14], etc. With millions and even billions of parameters, these deep models are prone to over-fitting, thus requiring regularization strategies to improve their generalization ability [33]. To tackle with over-fitting, many regularization techniques have been proposed, e.g., weight decay [32, 30, 28, 63], dropout [24, 60, 4, 62, 57], normalization [27, 54, 3, 26, 67], adding noise [25, 50], layer-wise pre-training and initialization [12, 21], label-smoothing [58], and so on. Among which, dropout and its variants are most popular owing to its effectiveness and moderate cost as well as good compatibility with other regularization methods [45], which has been successfully applied to regularize a wide range of neural network

architectures [49], e.g., convolutional neural network layers [64, 10], recurrent neural networks [17, 55, 42], Transformer [69, 75, 65]. The success of dropout methods can be interpreted by preventing co-adaptation of neurons and performing an implicit ensemble of sub models from dropout. Owing to the effect in promoting sparsity of weights and stochastic nature, dropout methods are also adapted to other applications, e.g., contrastive learning for sentence representation learning [18], neural network compression [44, 46] and model uncertainty estimation [16].

Unlike previous researches of designing specific dropout variants or adapting dropout to different applications, we consider to further regularize the model on the success of dropout. Specifically, any two sub models sampled from dropout are encouraged to produce consistent model prediction for an input data by utilizing KL-divergence in the training stage. That is, we conduct regularization on the model output level. In doing so, the sub model outputs produced by the randomness of dropout are regularized to reduce the parameter freedom, which will enhance generalization in inference.

**Consistency Training.** Besides regularization methods, our work also relates to a few works of consistency training on dropout models or data augmentation. Among them, the most representative methods are ELD [40], FD [77], and Cutoff [56]. As discussed in Section 2.4, ELD only focuses on the inconsistency between the sub model with dropout (train) and the expected full-model without dropout (inference), while FD works between the sub models only (consistence between two sub models). Both ELD and FD utilize $L_2$ to regularize the hidden space. Instead, our R-Drop performs consistency training on dropout from the output space with a more effective bidirectional KL loss. Unlike the above consistency training method on sub models, Cutoff resembles launching consistency training from a data perspective by regularizing the inconsistency between the original data the augmented samples with part of the information within an input sentence being erased.

**Self-distillation.** Minimizing the KL-divergence between the output distributions of two different models correlates with knowledge distillation [23, 15, 2, 35, 13, 74], where the two models refer to teacher and student, respectively. In our setting, the teacher and student are the dropout instantiations of the same model, and thus it resembles self-knowledge distillation [43] scenario. Different from existing method that exploits dark knowledge from the model itself [20, 19] or distills knowledge between different layers [71], our strategy can be regarded as an instance-wise self-knowledge distillation, i.e., each pair of sampled sub models perform distillation between each other for the same input, which also relates to mutual learning [72] but ours is much more efficient without extra parameters.

# 6  Conclusions and Future Work

In this paper, we proposed a simple yet very effective consistency training method built upon dropout, namely R-Drop, which minimizes the bidirectional KL-divergence of the output distributions of any pair of sub models sampled from dropout in model training. Experimental results on 18 popular deep learning datasets show that not only can our R-Drop effectively enhance strong models, e.g., ViT, BART, Roberta-large, but also work well on large-scale datasets and even achieve SOTA performances when combined with vanilla Transformer on WMT14 English→German and English→French translations. Due to the limitation of computational resources, for pre-training related tasks, we only tested R-Drop on downstream task fine-tuning in this work. We will test it on pre-training in the future. In this work, we focused on Transformer based models. We will apply R-Drop to other network architectures such as convolutional neural networks.

## Acknowledgments and Disclosure of Funding

We would like to thank the reviewers for their constructive comments. Juntao Li is the corresponding author. This work was supported by the National Science Foundation of China (NSFC No. 62036004).

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
