# Appendix for "R-Drop: Regularized Dropout for Neural Networks"

## A  Detailed Experimental Settings

We provide more detailed settings for the experiments of each task in this part.

### A.1  Neural Machine Translation

For all the NMT tasks, we use the public datasets from IWSLT competitions[1] and WMT competitions[2]. We tokenize all the datasets with byte-pair-encoding (BPE) [11] approach with the dictionary built jointly upon the source and target sentence pairs except the IWSLT17 En↔Zh translation dataset that is built separately. After tokenization, the resulted vocabularies for IWSLT datasets are near $10k$, while for WMT datasets, the vocabulary size is about $32k$.

To train the Transformer based NMT models, we use `transformer_iwslt_de_en` configuration for IWSLT translations, which has 6 layers in both encoder and decoder, embedding size $512$, feed-forward size $1,024$, attention heads 4, dropout value $0.3$, weight decay $0.0001$. For the WMT experiments, the `transformer_vaswani_wmt_en_de_big` setting has 6 layers in encoder and decoder, embedding size $1,024$, feed-forward size $4,096$, attention heads 16, dropout value $0.1$, attention dropout $0.1$ and relu dropout $0.1$. The training is optimized with Adam [5] with $\beta_1 = 0.9, \beta_2 = 0.98, \epsilon = 10^{-9}$. The learning rate scheduler is `inverse_sqrt` with default learning rate $0.0005$ and warmup steps $4,000$. Label smoothing [12] is adopted with value $0.1$. Our code implementation is based on open-source Fairseq[3]. We train the IWSLT translations on 1 GEFORCE RTX 3090 card and the WMT translations on 8 GEFORCE RTX 3090 cards.

To evaluate the performance, we use `multi-bleu.perl`[4] to evaluate IWSLT14 En↔De and all WMT tasks for a fair comparison with previous works [15, 9]. For other NMT tasks, we use `sacre-bleu`[5] [10] for evaluation. When inference, we follow [13] to use beam size 4 and length penalty $0.6$ for WMT14 En→De, beam size 5 and penalty $1.0$ for other tasks. We further report the `sacre-bleu` evaluated BLEU score on WMT14 En→De and En→Fr tasks to show advanced comparisons, where the corresponded results are $29.5$ and $41.8$, respectively.

### A.2  Abstrative Summarization

For summarization, we take the pre-trained BART [6] model as backbone and fine-tune on the CNN/DailyMail dataset[6]. BART is a pre-trained sequence-to-sequence model based on the masked source input and autoregressive target output, which contains 12 layers of Transformer encoder and 12 layers of Transformer decoder, the embedding size is $1,024$, and the feed-forward size is $4,096$. Dropout value is $0.1$. During fine-tuning, we follow the hyper-parameters used in [6]. The pre-trained

---

[1] https://iwslt.org/

[2] https://www.statmt.org/wmt14/translation-task.html

[3] https://github.com/pytorch/fairseq/tree/master/examples/translation

[4] https://github.com/moses-smt/mosesdecoder/blob/master/scripts/generic/multi-bleu.perl

[5] https://github.com/mjpost/sacrebleu

[6] https://github.com/abisee/cnn-dailymail

| Hyper-parameter | CoLA | MRPC | RTE | SST-2 | MNLI | QNLI | QQP | STS-B |
|---|---|---|---|---|---|---|---|---|
| Learning Rate | 1e-5 | 1e-5 | 1e-5 | 1e-5 | 1e-5 | 1e-5 | 1e-5 | 1e-5 |
| Max Update | 5336 | 2296 | 3120 | 20935 | 123873 | 33112 | 113272 | 3598 |
| Max Sentence (Batch) | 16 | 16 | 8 | 32 | 32 | 32 | 32 | 16 |
| Dropout | 0.1 | 0.1 | 0.1 | 0.1 | 0.1 | 0.1 | 0.1 | 0.1 |
| Coefficient $\alpha$ | 0.5 | 1.0 | 1.0 | 1.0 | 0.5 | 1.0 | 0.5 | 1.0 |

Table 1: Hyper-parameters when fine-tuning our models on GLUE benchmark.

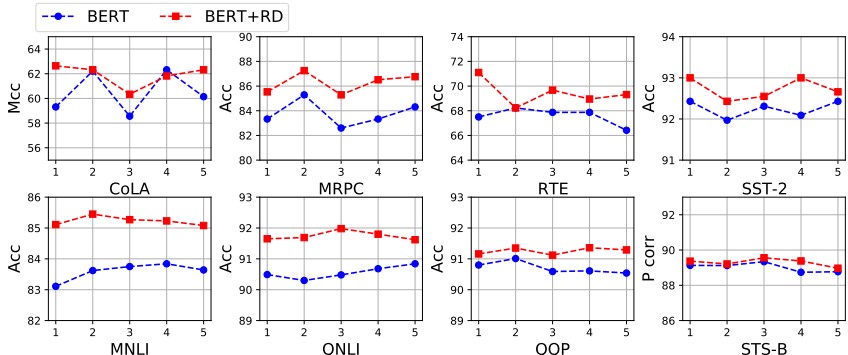

Figure 1: Results on 8 GLUE tasks with different random seeds.

model and the backbone implementations are all from Fairseq[7]. The training is conducted on 8 GEFORCE RTX 3090 GPU cards.

## A.3 Language Modeling

For language modeling, we train on the Transformer decoder [13] and Adaptive Input Transformer [1] models. The configuration for Transformer is `transformer_lm_gpt`, which contains 12 layers with embedding size 768 and feed-forward size 3,072, attention heads 12. Dropout and attention dropout are 0.1. For Adaptive Input Transformer, the configuration is `transformer_lm_wiki103` with 16 layers, embedding size 1,024, feed-forward size 4,096, attention heads 16, dropout 0.3, attention dropout 0.1, gelu dropout 0.1 and adaptive softmax dropout 0.2. We train Transformer model for $50k$ steps and Adaptive Input Transformer for $286k$ steps. The development is based on the code base Fairseq[8]. The training is on 8 Tesla V100 GPU cards.

## A.4 Language Understanding

For language understanding tasks, we follow the popular pre-training and fine-tuning methodology, and the fine-tuned sets are the GLUE [14] benchmark. We follow previous works [2, 7] to work on the 8 tasks, including singe-sentence classification tasks (CoLA, SST-2), sentence-pair classification tasks (MNLI, QNLI, RTE, QQP, MRPC), and the sentence-pair regression task (STS-B). The detailed data statistics can be found from the original paper [14].

The pre-trained BERT-base model is the Transformer [13] encoder network, which contains 12 layers with embedding size 768, feed-forward size 3,072 and attention heads 12. Correspondingly, the Roberta-large model contains 24 layers with embedding size 1,024, feed-forward size 4,096 and attention heads 16. During fine-tuning, we use Adam [5] as our optimizer with $\beta_1 = 0.9$, $\beta_2 = 0.98$, $\epsilon = 10^{-6}$, and $L_2$ weight decay of 0.01. We select the learning rate in range $\{5 \times 10^{-6}, 10^{-5}\}$ and batch size in $\{8, 16, 32\}$. Other hyper-parameter settings are mostly same as previous works [7]. The pre-trained model and the backbone implementations are all from Huggingface Transformers[9]. We

---

[7] https://github.com/pytorch/fairseq/tree/master/examples/bart
[8] https://github.com/pytorch/fairseq/tree/master/examples/language_model
[9] https://github.com/huggingface/transformers

| $\alpha$ | 0.1 | 0.5 | 1.0 | 1.5 |
|------|-------|-------|-------|-------|
| **MRPC** | 84.30 | 86.03 | **86.51** | 85.78 |
| **SST-2** | 92.54 | 92.77 | **93.02** | 92.43 |
| **MNLI** | 84.20 | **84.48** | 83.44 | 82.27 |
| **QNLI** | 91.21 | 91.92 | **92.01** | 91.12 |

Table 2: Comparison of the effect of different $\alpha$ for some GLUE tasks.

report the specific settings of several important hyper-parameters in Table 1, including the dropout value. The fine-tuning experiments are conducted on 1 GEFORCE RTX 3090 GPU card.

Further, to give a clear comparison of our R-Drop based fine-tuning and vanilla fine-tuning, we plot the performance changes from different random seeds over the pre-trained BERT model on each GLUE task. The curves are shown in Figure 1. We can see that consistent improvements are achieved on different random seeds, which means our R-Drop can robustly help improve the model generalization and model performance. Besides, we also provide some task performances of different $\alpha$ values, shown in Table 2. This result demonstrated that $\alpha$ indeed is a sensitive hyper-parameter for each GLUE task, while $\alpha = 1.0$ is a good choice for most tasks.

**MSE Regularization**    Our R-Drop is presented under the KL-divergence between two distributions. To extend our method into the regression task, such as STS-B in GLUE, we introduce the MSE-based regularization. For input data $(x, y)$, we forward the $x$ two times similarly as in classification and obtain the two predicted values $y_1'$ and $y_2'$. Then we regularize these two predicted values with MSE as follow:

$$\mathcal{L}_{mse_r} = ||y_1' - y_2'||_2, \tag{1}$$

and we add $\mathcal{L}_{mse_r}$ with conventional MSE loss: $\mathcal{L}_{mse} = ||y - y_1'||_2 + ||y - y_2'||_2$. The final optimization objective is:

$$\mathcal{L} = \mathcal{L}_{mse} + \alpha \mathcal{L}_{mse_r}. \tag{2}$$

### A.5   Image Classification

The image classification task is evaluated with the recent popular Vision Transformer (ViT) [4] model, which is the same as Transformer but with the image patch data as input. We take the two publicly released models[10], ViT-B/16 and ViT-L/16, which are pre-trained on ImageNet-21k [3] dataset with $21k$ classes and $14M$ images in total. ViT-B/16 is a Transformer model with 12 Transformer encoder layers, embedding size 768, feed-forward size $3,072$ and attention heads 12, while ViT-L/16 with 24 layers, $1,024$ embedding size, $4,096$ feed-forward size and 16 attention heads. We only conduct the fine-tuning stage experiments on CIFAR-100 and ImageNet. Note that the ImageNet results are computed without additional techniques (Polyak averaging and 512 resolution images) used to achieve results in [4]. During fine-tuning, the dropout values are $0.1$ for both models. Fine-tuning is on 8 GEFORCE RTX 3090 GPU cards.

## B   Theoretical Discussion of R-Drop

In this section, we provide the proof for Proposition 2.1 in the main paper and make some discussions.

**Proposition B.1.** *For a linear model $\mathcal{P}^w(y|x) = \mathtt{softmax}(\mathtt{Norm}(w^T x))$ where $\mathtt{Norm}(\cdot)$ denotes the normalization layer and $x \in \mathbb{R}^d$, with the constraint in Equation (6) in the main paper, we have $|\mathcal{L}_{nll}(w) - \mathbb{E}_\xi[\mathcal{L}_{nll}(w, \xi)]| \leq c\sqrt{\epsilon}$, where $\mathcal{L}_{nll}(w), \mathcal{L}_{nll}(w, \xi)$ are the empirical loss calculated by the full model and a random sub model respectively, $c$ is a constant related to the Liptschtz constant of the softmax operator.*

---

[10]`https://github.com/jeonsworld/ViT-pytorch`

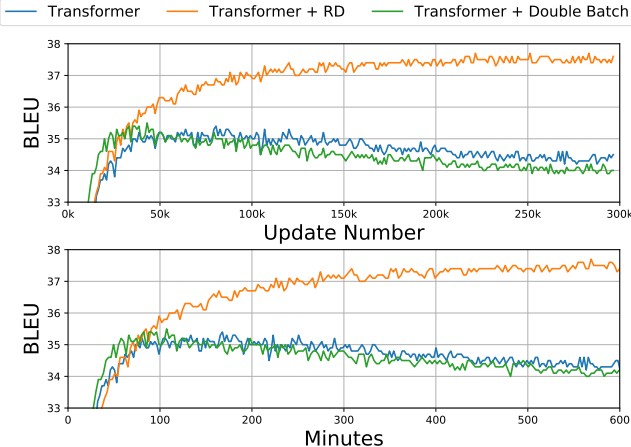

Figure 2: Results of R-Drop and Transformer with a doubled batch size.

*Proof:* Here, the normalization operator normalize the row of the weight matrix $w$ to be 1. According to the Lipschitz continuity of the loss, we have

$$|\mathcal{L}_{nll}(w) - \mathbb{E}_{\xi}[\mathcal{L}_{nll}(w, \xi)]| \le c_1 \cdot \frac{1}{n} \sum_{i=1}^{n} \mathbb{E}_{\xi} \|w^T x_i - \frac{1}{p} \cdot (w^T x_i) \odot \xi\| \tag{3}$$

$$= c_1 \cdot \frac{1}{n} \sum_{i=1}^{n} (1-p) \|w^T x_i\| \tag{4}$$

where $c_1$ is the Lipshitz constant.
According to the condition $\frac{1}{n} \sum_{i=1}^{n} \mathbb{E}_{\xi^{(1)}, \xi^{(2)}} [\mathcal{D}_{KL}(\mathcal{P}^w_{\xi^{(1)}}(y_i|x_i) || \mathcal{P}^w_{\xi^{(2)}}(y_i|x_i)))] \le \epsilon$ , we have

$$\frac{1}{2n} \sum_{i=1}^{n} \mathbb{E}_{\xi^{(1)}, \xi^{(2)}} \|\mathcal{P}^w_{\xi^{(1)}}(y_i|x_i) - \mathcal{P}^w_{\xi^{(2)}}(y_i|x_i))\|_1 \le \sqrt{\frac{\epsilon}{2}}, \tag{5}$$

because the relation between the KL-divergence and the total variation distance. Since we constrain the norm of $w$, for fixed $x$, the softmax operator is a bijection from $w^T x$ to $\texttt{softmax}(w^T x)$. Suppose $c_2$ is the Lipschitz constant of the inverse function from $\texttt{softmax}(w^T x)$ to $w^T x$, we have

$$\frac{1}{2n} \sum_{i=1}^{n} \mathbb{E}_{\xi^{(1)}, \xi^{(2)}} \|\frac{1}{p} \cdot w^T x_i \odot \xi_1 - \frac{1}{p} \cdot w^T x_i \odot \xi_2\| \le c_2 \sqrt{\frac{\epsilon}{2}} \tag{6}$$

For the left term in Eq.(5), we have $\frac{1}{2np} \sum_{i=1}^{n} \mathbb{E}_{\xi^{(1)}, \xi^{(2)}} \|w^T x_i \odot \xi_1 - w^T x_i \odot \xi_2\| = \frac{1-p}{n} \sum_{i=1}^{n} \|w^T x_i\|$, because $\xi^{(1)}, \xi^{(2)}$ independently follow Bernoulli distribution. Then we have $\frac{1}{n} \sum_{i=1}^{n} \|w^T x_i\| \le \frac{c_2}{1-p} \sqrt{\frac{\epsilon}{2}}$. Combined with Eq.(4), we have

$$|\mathcal{L}_{nll}(w) - \mathbb{E}_{\xi}[\mathcal{L}_{nll}(w, \xi)]| \le c_1 c_2 \sqrt{\frac{\epsilon}{2}} \tag{7}$$

Let $c = \sqrt{\frac{1}{2}} c_1 c_2$, we can get the result.

## C   More Studies

### C.1   Batch Size Doubled Training

As discussed in Section 2.2, we implement the algorithm by repeating input data $x$ once and concatenating the $x$ with repeated one in the same mini-batch to forward once. This is similar to enlarging the batch size to be double at each step. The difference is that half of the data are the

| Minutes | 10min | 30min | 60min | 90min | 150min | 200min | 300min |
|---------|-------|-------|-------|-------|--------|--------|--------|
| **Baseline** | 27.14 | 33.62 | 34.18 | 34.54 | - | - | - |
| **R-Drop** | 16.06 | 31.56 | 33.61 | 34.91 | 35.74 | 36.13 | 36.39 |

Table 3: Comparison Baseline and R-Drop (reduced half-batch training) BLEU score along with training time on IWSLT14 De→En translation.

| Model | Acc (CIFAR-100) | BLEU (IWSLT14 De→En) |
|-------|-----------------|----------------------|
| Baseline | 77.1 | 34.78 |
| FD [16] | 77.6 | 35.04 |
| **R-Drop** | 78.13 | 37.25 |

Table 4: Comparison of Baseline, FD and R-Drop on IWSLT14 De→En and CIFAR-100 tasks.

same as the other half, while directly doubling the batch size, the data in the same mini-batch are all different. Therefore, we are still interested in the result of directly doubling the batch size to see the performance. We conduct experiments on IWSLT14 De→En translation with Transformer, and the batch size is enlarged from $4,096$ to be $8,192$. The result is $34.93$ BLEU score. We can see that though slight improvement is achieved (compared to baseline $34.64$), it falls far behind our strong performance $37.25$. For the detailed training cost for each step, we present the number here: Transformer + Double Batch costs near 9ms per step, while Transformer + DR costs about 10ms per step. The additional cost is from the KL-divergence loss backward computation. We can see the cost is $1.1$ times, which is a negligible cost. We also plot the valid BLEU curves along with the training for this study. The curves are shown in Figure 2. Compared to this batch size doubled training and our R-Drop, we can clearly see the advantage of R-Drop. With similar training costs, R-Drop gradually improves the performance to a much stronger one. In the figures, we also plot the curve for Transformer with the original batch size training (e.g., $4,096$) for a better comparison.

## C.2 Importance of KL-Divergence

Our method introduces a KL-divergence between the two distributions from the same sample. In this study, we specifically investigate the importance of KL-divergence loss. Thus, this ablation removes the $\mathcal{L}_{KL}$ loss between the two distributions and only keeps the $\mathcal{L}_{NLL}$ loss for training. Similar to other studies, we work on IWSLT14 De→En translation, and the model is Transformer. The result is also $34.93$ BLEU score (same as above enlarged batch size), which is slightly better than the Transformer baseline ($34.64$), but far worse from our R-Drop based training result $37.25$ BLEU score. This result well demonstrates the importance and effectiveness of our introduced KL-divergence loss.

## C.3 Training Time and Efficiency

To fairly compare the training time between baseline and R-Drop model for the same batch size, We reduce the batch size for the R-Drop model and provide the BLEU score along with training time on IWSLT14 De→En translation tasks, which is shown in Table 3. From the table, we can see that though R-Drop needs more training to converge, the final model is much better than the baseline. We can see that the time cost is comparable for reduced half-batch training when reaching the BLEU score equal to the baseline.

## C.4 Experiments for ELD and FD

To give a better understanding of the advantages of R-Drop over hidden space regularization, we also conduct experimental studies. Since FD [16] has shown its advantage over ELD [8], we mainly present experimental comparisons between FD and R-Drop. The experiments are image classification on the cifar-10 dataset (used in FD work) and IWSLT14 De→En Translation (used in R-Drop work). The cifar10 experiments are conducted on the released code[11], and we add R-Drop regularization as our implementation. For the image classification task, the R-Drop model hyper-parameter $\alpha$ needs to reduce linearly with the learning rate implemented by `torch.optim.lr_scheduler.MultiStepLR`.

---

[11]https://github.com/akamaster/pytorch_resnet_cifar10

The IWSLT experiments are on our released code, and we implement the FD same as its released code. From Table 4, we can clearly see that R-Drop is superior to FD on both tasks, which can prove the advantages of the KL-divergence consistency regularization.

## C.5  Ensemble and Weight Averaging

| Model Number
Model Settings | 1 | 2 | 3 | 4 | 5 | 6 |
|---|---|---|---|---|---|---|
| Transformer full-model deep ensembling | 34.78 | 36.06 | 36.37 | 36.59 | 36.80 | 36.92 |
| Transformer full-model weight averaging (model) | 34.78 | 0 | 0 | 0 | 0 | 0 |
| Transformer full-model weight averaging (checkpoint) | 34.78 | 35.00 | 35.21 | 35.26 | 35.29 | 35.44 |
| R-Drop full-model weight averaging (checkpoint) | 37.25 | 37.22 | 37.27 | 37.30 | 37.22 | 37.31 |

Table 5: Comparison of BLEU scores achieved by model deep ensembling or weight averaging with different model (trained with different random seed) and epoch checkpoint (trained with same seed).

We also compare our R-Drop based training with deep ensembling and weight average methods. (1) Deep ensembling usually utilizes multiple models with different parameters, which incurs additional inference costs, including both numbers of model parameters and inference time. (2) Parameter/weight averaging directly averages the multiple model parameters, and it can only be useful when the parameters are not far away from each other. In contrast, R-Drop aims to make sub models consistent within dropout-based training. It is simple yet effective by adding a consistent regularization loss without any other modifications. The model parameters are not increased, and the inference cost is the same as a single model. We train several independent models with different parameters (each with dropout) and then do deep ensembling, weight averaging on these full models. The results are show in Table 5 . Obviously, R-Drop achieves great performance with a single full model (e.g., 37.25), much better than weight averaging and deep ensembling. Ensemble methods improve the independently trained full models, but the best result is still from R-Drop (at least with 6 models ensembling here). Weight averaging can improve the performance of the base dropout model by averaging nearby checkpoints with the same random seed, but the improvement is relatively small compared with R-Drop. Besides, the weights average obtains an extremely low result for different random seeds trained models (BLEU=0) due to the far difference of model parameters.