# OpenReview forum: "R-Drop: Regularized Dropout for Neural Networks"
_NeurIPS.cc/2021/Conference — NeurIPS 2021 Poster_

### Official Review · Reviewer_fC6N · 2021-07-15

**Rating:** 7
**Confidence:** 3

**Summary:**

The paper introduces a variant of dropout called R-drop. First, the data goes through the forward pass once, and we obtain the output distribution. Next, the data goes through the forward pass (with the same dropout rate but different dropout neurons) another time, and we obtain the output distribution. The goal is to minimize the loss, which is the standard negative log-likelihood (adding two NLL losses from two different passes together) together with KL divergence (KL(p1 || p2) + KL(p2 || p1)). There is large performance gain on machine translation, GLUE tasks, as well as language modeling.

**Limitations And Societal Impact:**

I don't see potential negative societal impacts.

**Main Review:**

Strengths

- Elegant method and easy implementation for practitioners.

- Evaluated on many tasks. Hyperparameter information is comprehensive in the main text and the appendix.

- Theoretical analysis in the appendix.


Comments / points for improvements

My only major issue about the paper is as follows. R-drop somehow regularizes the model parameters so that they are not too extreme. (1) Parameter averaging and (2) deep ensembling also effectively try to make sure that the parameters / output distributions are not too extreme.

For parameter averaging, one can average the parameters of two trained networks (and perhaps finetune it a little). For deep ensembling, we simply predict two output distributions (using two trained networks) and then take the average of the distributions. Do the authors think that these are relevant baselines?

How would the authors think R-drop would perform compared to those baselines? What are the pros and cons of R-drop compared to deep ensembling for example?


Another question: The alpha value (weight for KL divergence term) seems to be a sensitive hyperparameter. How sensitive is alpha for GLUE tasks / language modeling?


**Time Spent Reviewing:**

2-2.5

---

> ### Author Response · Authors · 2021-08-10
> **Response to Reviewer fC6N**
>
> Thanks for your review comments!
>
> >Q. Comparison with deep ensembling and parameter averaging.
>
> A: R-Drop aims to make sub models consistent within dropout based training. It is simple yet effective by adding a consistent regularization loss, without any other modifications. The model parameters are not increased, and the inference cost is the same as a single model. Deep ensembling usually utilizes multiple models with different parameters, which incurs additional inference costs, including both numbers of model parameters and inference time. Parameter averaging can only be useful when the parameters are not far away from each other (see the below experiments). Following your suggestion, we compare experimental results of parameter average, deep ensembling, and R-Drop. We train several independent models with different parameters (each with dropout) and then doing parameter average, deep ensembling on these full models. The results are:
>
>
> **IWSLT14 German->English translation**
>
> | Setting\models | 1 | 2 | 3 | 4 | 5 | 6 |
> | --------- |  ---- | ---- | ---- | ---- | ---- | ---- |
> | Base full model ensemble  | 34.78 | 36.06 | 36.37 | 36.59 | 36.80 | 36.92 |
> | Base full model weight averaging (different seed)  | 34.78 | 0 | 0 | 0 | 0 | 0 |
> | Base full model weight averaging (same seed) | 34.78 | 35.00 | 35.21 | 35.26 | 35.29 | 35.44 |
> | R-Drop full-model weight averaging (same seed)  | 37.25 | 37.22 | 37.27 | 37.30 | 37.22 | 37.31 |
>
> Obviously, R-Drop achieves great performance with a single full model (e.g., 37.25), much better than weight averaging and deep ensembling.
> Ensemble methods improve the independently trained full models, but the best result is still from R-Drop (at least with 6 models ensembling here). Weight averaging can improve the performance of base dropout model by averaging nearby checkpoints with the same random seed, but the improvement is relatively small compared with R-Drop. Besides, the weight average obtains an extremely low result for different random seeds trained models (BLEU=0) due to the far difference of model parameters.
>
> >Q. As for the $\alpha$ sensitivity for GLUE tasks.
>
> A: We show on MPRC/SST-2/MNLI/QNLI tasks as follows:
>
> **MRPC**
>
> | $\alpha$ | accuracy | F1 |
> | -------  | ---- | ---- |
> | 0.1 | 84.30 | 89.10 |
> | 0.5 | 86.03 | 90.05 |
> | 1.0 | 86.51 | 90.96 |
> | 1.5 | 85.78 | 90.20 |
> | 2.0 | 85.29 | 89.93 |
>
> **SST-2/MNLI/QNLI**
>
> | $\alpha$ | SST-2 | MNLI | QNLI |
> | -------  | ---- | ---- | ---- |
> | 0.1 | 92.54 | 84.20 | 91.21 |
> | 0.5 | 92.77 | 84.48 | 91.92 |
> | 1.0 | 93.02 | 83.44 | 92.01 |
> | 1.5 | 92.43 | 82.27 | 91.12 |
>
> As can be seen, $\alpha$ is indeed an important hyperparameter. While the impact of $\alpha$ is different across tasks, overall, $\alpha=1.0$ is good choice for most tasks.

---

> ### Comment · Reviewer_fC6N · 2021-09-02
> **Thank you authors**
>
> I've been following the discussion. The author's responses are convincing. It's very very interesting that R-drop does better than ensembling.
>
> The summary of the revision plan also looks convincing.

---

### Official Review · Reviewer_Vd3y · 2021-07-16

**Rating:** 7
**Confidence:** 4

**Summary:**

This paper proposes a simple yet effective regularization method for deep learning models. The model performs two forward passes with different dropout masks for the same input and minimizes the difference between the two output distributions by the bidirectional KL divergence. The proposed method has been sufficiently validated on both CV and NLP benchmarks.

**Limitations And Societal Impact:**

Please refer to the weaknesses.

**Main Review:**

### Update after Response:
Thank you for the very detailed response, I would like to increase my score to 7 since some of my concerns have been alleviated.

### Strengths:

1.	The proposed method is simple and might be widely used in the community.
2.	The empirical results are strong. I ran the experiment on the IWSLT German-English translation task and it can indeed achieve the BLEU scores shown in the paper.
3.	The code is released and the documentation is well-organized.
4.	The experiments are conducted on both NLP and CV tasks, showing the strong generality of the proposed method.

### Weaknesses:
1. (**Main**) The proposed method can be seen as a combination of [cutoff](https://arxiv.org/abs/2009.13818) (very similar) and [simCSE](https://arxiv.org/abs/2104.08821), but this paper does not cite and discuss them. Is the proposed method inspired by them? What is the relationship between these methods?

2.	The authors should report sacreBLEU scores for machine translation experiments to improve the reproducibility, especially for Table 2 where I find the results are not comparable since the models use different tokenizations. Could you provide the scores in the rebuttal? **after response**: I don't agree with the authors that the results are comparable. [sacreBLEU paper](https://aclanthology.org/W18-6319/) has clearly shown that BLEU scores produced by multi-bleu.perl are hard to compare and this is the reason why they proposed sacreBLEU score for a fair comparison. Furthermore, as the scareBLEU score of R-Drop is much lower than Transformer Admin, I suggest the authors remove the claim of SOTA results on the En-De task.

3.	I find the proposed method is of low training efficiency (the small IWSLT task needs to run over 200 epochs). Could you discuss it more and provide the wall time of training in the rebuttal and the future paper?

4.	The paper does not provide any qualitative analyses. It is not clear that how the proposed method improves the models? For example, does the proposed method improve the translation of rare words in machine translation experiments? It would be nice if the authors could provide some qualitative analyses, which can provide more insights for future research.

5.	How does the $m$-time ($m$>2) drop deal with the bidirectional KL divergence? Could you give the formulations? Could you provide the 3-time result on the WMT English-German task? The claim in this part is somewhat arbitrary.

6.	I think the authors should not update the code after the supplementary material submission deadline. This might hurt the fairness of the community since the other submissions have no chance to update.



**Time Spent Reviewing:**

10

---

> ### Author Response · Authors · 2021-08-10
> **Response to Reviewer Vd3y**
>
> Thanks for your review comments and acknowledgement, especially for reproducing the experiment results with our released code!
>
> We provide the following answers for your concerns.
>
>
> >Q1.  As for the related work cutoff and SimCSE.
>
> A1: Both of them are motivated from the data augmentation perspective. The differences between them and our R-Drop are: (1) cutoff erases part of the information within an input sentence to generate the augmented samples, and the goal is to endow the learned representations with stronger generalization ability. In contrast, R-Drop focuses on the consistency between sub models for dropout based training and operates on the sub-model level instead of data level. Besides, R-Drop simply goes forward the input data twice for training, while cutoff incorporates multiple augmented samples for each sample during training. (2) Similarly, SimCSE aims at providing a positive example with an efficient implementation in the contrastive learning scenario. Hence, SimCSE views the two dropout samples as two augmented samples. There are no concepts like positive or negative samples in R-Drop. Besides, the training objective/loss of SimCSE is the contrastive loss based on the cosine similarity of hidden states, while R-Drop incorporates KL-divergence between output distributions between two sub models instead of hidden states, which is more suitable for measuring the model consistency (KL-divergence also eases the optimization). Furthermore, the role of the loss function is also different: the KL-divergence loss plays as a regularization term instead of the main training objective as in SimCSE.
>
> We want to point out that SimCSE and R-Drop are two contemporary works. We are not inspired by SimCSE. SimCSE is relased on Arxiv on 18 Apr, while we are busy working on our experiments that time (5 different tasks and 18 datasets, and many of them are large datasets) for NeurIPS submission deadline, 28 May. The gap between two works is only one month. We will add those discussions in the new version according to your suggestion.
>
> >Q2. The BLEU results in Table 2.
>
> A2: Thanks for the comment. We’d like to clarify that the results in Table 2 are indeed comparable. First, the tokenizations and the pre-processing scripts are same as provided in fairseq (https://github.com/pytorch/fairseq/blob/master/examples/scaling_nmt/README.md), and the works presented in Table 2 all utilize the fairseq toolkit (e.g., https://github.com/nxphi47/data_diversification, https://github.com/bert-nmt/bert-nmt). Second, the results in Table 2 are all evaluated by multi-bleu.perl, which are reported in their papers (also from https://paperswithcode.com/sota/machine-translation-on-wmt2014-english-german), and most of them only have the multi-bleu.perl evaluated BLEU scores (except Transformer Admin). We follow the same process as in BERT-fused NMT, Data Diversification to report BLEU scores measured by the multi-bleu.perl. For your suggestion, we also report the sacreBLEU score, which are 29.5 on WMT14 En->De and 41.8 on WMT14 En->Fr. Again, please note that multi-bleu.perl is more commonly utilized in the community. --- As for the results in Transformer Adamin (https://arxiv.org/pdf/2008.07772.pdf), they report multi-bleu.perl evaluated BLEU score 30.1 and sacreBLEU 29.5, R-Drop is not much lower.
>
> >Q3. Training time and efficiency.
>
> A3: As we discussed in section 4.1, the training cost for each step and the convergence time indeed increase. This is same as other regularization methods (e.g., larger dropout value takes more time to converge). As shown in section 4.1, though R-Drop needs more training to converge, the final model is much better than the baseline. Following your suggestion, we provide the wall time for 100k step training on IWSLT14 De->En translation, which is 9901s for baseline model and 9502s for R-Drop reduced half batch training, 19069s for R-Drop double batch training. We can see that the time cost is comparable when we use reduced half-batch training. Furthermore, the BLEU scores over the test set w.r.t the training time are as follows (reduced half-batch training).
>
> **IWSLT14 German->English translation**
>
> | Minutes | 10min | 30min | 60min | 90min| 150min | 200min | 300min |
> | --------- |  ---- | ---- | ---- | ---- | ---- | ---- | ---- |
> | baseline  | 27.14 | 33.62 | 34.18 | 34.54 | 34.55 |- | - |
> | R-Drop    | 16.06 | 31.56 | 33.61 |34.91 | 35.74 |36.13 | 36.39 |
>
> For large WMT experiments, we simply run about 2 days on A100 during training. We believe that continue training can further improve the BLEU scores.
>
> >Q4. For qualitative analysis.
>
> A4: R-Drop is a general method that can be applied to any dropout based neural network training, instead of domain specific problems such as in machine translation or language modeling. R-Drop studies the regularization effect (overfitting problem) and the sub-model consistency in dropout training. As analyzed in section 4.1, the sub models indeed become more consistent.
>
> We show several cases below about the rare word translations as you suggested.
>
> **IWSLT14 German->English translation case study**
>
> |  | sentence |
> | --------- |  ---- |
> | source  | aber die sache mit tryptaminen ist die , dass sie nicht oral eingenommen werden können , da sie durch ein im menschlichen darm natürlich vorkommendes enzym mit namen monoamin-oxidase , denaturiert werden. |
> | target  | but the thing about tryptamines is they cannot be taken orally because they're denatured by an enzyme found naturally in the human gut called monoamine oxidase . |
> | baseline translation  | but the thing about tryptamins is that they can't be taken in oral because , of course , through a human gut , they're named monoamine oxidase . |
> | R-Drop translation | but the thing about tryptamins is that they can't be taken in , because they're taken by an enzyme in the human gut , of course , called monoamine oxidase , bf denatured. |
>
> We can see that the rare words are indeed translated with better accuracy. In particular, the word "enzym" and "denaturiert werden" in the source language (De) was correctly translated into "enzyme" and "denatured" in the target language (En) by the R-Drop model. It is worth noting that "enzym" and "denatured" are professional biological words, and also rare words in the corpus.
>
> >Q5. For the m-time drop.
>
> A5: We can simply take the bidirectional KL-divergence between each pair of two sub models. The formulation is: $L_{KL} = \frac{\alpha}{m*(m-1)}\sum_{\{i,j\}\in\{1,...,m\}, i!=j}[D_{KL}(P_i^w(y|x) || P_j^w(y|x))]$. As you suggested, we run 3-time drop on WMT14 En->De, and the result is 30.89 BLEU score, which is similar to 2-drop result. Therefore, this again shows that the 2-drop is already a strong regularization.
>
> >Q6. The code update.
>
> A6: We agree with you about code update. We want to clarify that none of the important part of the algorithm is changed. We modified mostly on the readme file (e.g., fixing typos), which does not change the results and impact reproduction. We will be more cautious in the future and thanks for your advice!
>
> Hope the above replies can answer your questions.

---

> > ### Comment · Area_Chair_VDVh · 2021-08-25
> > **Re: Response to Reviewer a8g8**
> >
> > Dear Reviewer a8g8,
> > I want to thank you for your diligent review and want to second the authors gratefulness for running their code and reproducing results.
> >
> > Does the authors' detailed response change your assessment ?
> >
> > As a side point to the authors, I would hope that the changes to code after submissions are not methodological changes. Code release is not mandatory of course (but really appreciated since others can reproduce results, as a8g8 did above), but since the goal is to let other reproduce your methods I would hope any changes are purely cosmetic.

---

> > > ### Author Response · Authors · 2021-08-28
> > > **Thanks!**
> > >
> > > Dear AC and Reviewer Vd3y,
> > >
> > > Thanks for your comments/suggestions again, we promise the changes are not methodological changes. We will be more cautious in the future. Appreciate your time a lot.

---

### Official Review · Reviewer_a8g8 · 2021-07-16

**Rating:** 5
**Confidence:** 3

**Summary:**

The paper proposes the dropout-based regularization method R-Drop. The proposed R-Drop further regularizes the output predictions of sub-models from dropout.  The theoretical analysis was performed to show that the proposed method reduces the freedom of model parameters. The empirical results on multiple datasets were performed to show that the proposed method improves the performance (accuracy) for four NLP tasks and one CV task.

**Limitations And Societal Impact:**

The limitation of the current approach was mentioned in the paper (in line 103). Is there any way this limitation can be mitigated?

**Main Review:**

The method needs clarity. For example,

Re: "Concretely, given the input data xi at each training step, we feed xi to go through the forward pass  of the network twice"

Does each forward pass change parameters involved in both distributions (P1w (yi |xi ) and P2w (yi|xi )) learning?

Concretely, it will be useful to elaborate following steps of the algorithm:
4:  forward the data one time and obtain the output distribution P1w (yi |xi ),
5:  forward the data one more time and obtain the output distribution P2w (yi |xi ),

Interestingly, at the implementation, there were not actually two passes, rather input concatenation was done resulting in larger batch size to reduce the time complexity (as per lines 100-101). Why not keep just one type of formulation of the method (i.e. two passes or larger batch size)?

The model parameters are increased in the proposed methods, and potentially converge times increases to learn extra parameters. I am not sure the performance improvement is due to the better regularization or larger model with longer training.



**Time Spent Reviewing:**

4 hours

---

> ### Author Response · Authors · 2021-08-10
> **Response to Reviewer  a8g8**
>
> Thanks for your review comments. Seems you have some misunderstandings about our paper.
>
>
> >Q: Does each forward pass change parameters involved in both distributions $P_1^w (y_i |x_i)$ and $P_2^w (y_i|x_i))$ learning?
>
> A: We think the algorithm has clearly illustrated the model updating process. As described in algorithm line 4 and 5, we forward a data sample twice, and line 8 shows we update the model only once. That is, after the data sample is loaded in memory: `x = dataloader(train_set)`, it will go through the forward pass twice to obtain $P_1$ and $P_2$, $P_1$ = `model.forward(x)`, $P_2$=`model.forward(x)`, and then go backward once, `model.backward(`$P_1$, $P_2$`)`. The model parameters are not updated during the two forward passes, and only updated once in the backward pass (see algorithm line 8).
>
> >Q: Why not keep just one type of formulation of the method (i.e. two passes or larger batch size)?
>
> A: As stated in Line 100, the double batch training aims to reduce the training cost since the two forward passes take more time than one forward pass. In implementations, we only utilize the double batch training considering the efficiency (you can check our github codes). The two-pass algorithm is shown for a better and easier understanding of our method. We will make this clearer in the new version.
>
> >Q: The model parameters are increased in the proposed methods, and potentially converge times increases to learn extra parameters.
>
> A: This is a **misunderstanding** of our method. R-Drop only adds a regularization loss without changing any part of the model; therefore, no model parameters are increased in our method. The model parameters are exactly the same as the baseline models (e.g., default Transformer model). Besides, the increase of convergence time is due to the regularization, and we have discussed in section 4.1 of the regularization effect. As shown in Figure 2 (the right figures), during training, R-Drop quickly surpasses the baseline model and the accuracy of the final converged model is much better than baseline. In the last section of our supplementary, we have also shown the importance of regularization loss. Therefore, performance improvement is clearly due to the better regularization effect.

---

> > ### Comment · Area_Chair_VDVh · 2021-08-25
> > **Re: Response to Reviewer a8g8**
> >
> > Dear Reviewer a8g8,
> > it looks to me that the authors have addressed your questions. Could you please let us know if this affects your rating ?
> > Much appreciated.
> > Thanks

---

> > ### Comment · Reviewer_a8g8 · 2021-09-01
> > **Re: Response to Reviewer a8g8**
> >
> > Thanks for the responses. The response clarifies Q1. To be able to run the backward pass once on both weight sets (w) of the forward passes, the weight instances (w associated with the first pass and w associated with the second pass) will need to be stored for the backward pass if implemented with this approach. Otherwise, the second forward pass will overwrite w what was created in the first pass.
> >
> > However, responses to Q2 and Q3 are still not clear.
> >
> > It is clear to me that a large batch was used to improve computational cost and two passes were written in the algorithm to improve reader understanding. However, two approaches (i.e., two passes and a large batch) are not interchangeable, although both approaches will increase the computational cost, the larger batch size will require an increase in the model parameters (i will explain this in the next paragraph) and the two passes approach will require passing the inputs two times and storing w for the first pass to compute loss.
> >
> > A large batch will require a larger projection matrix (e.g. if x is of size p x q then the larger batch will be p x 2q) and that increases the computational (training) cost due to the larger projection matrix (which is the part of the model parameters). In the response, the authors mentioned that "R-Drop only adds a regularization loss without changing any part of the model; therefore, no model parameters are increased in our method." However, the regularization loss depends on one of the two approaches that increase the size of the parameter matrix. Moreover, in the paper, on pages 100-104, authors mentioned that "For implementation, to save the training cost, we do not forward the data twice, instead, we repeat the input x once and concatenate them ([x; x]) in the same mini-batch to forward once. Compared to the conventional training, our implementation is similar to enlarge the batch size to be double, and one potential limitation is that the computational cost of R-Drop increases at each step."
> >
> > I am inclining to keep the score unchanged.

---

> > > ### Author Response · Authors · 2021-09-01
> > > **Re: Re: Response to Reviewer a8g8**
> > >
> > > Dear Reviewer a8g8,
> > >
> > > Thanks for your post-response. We would like to discuss more with you about the questions.
> > >
> > > > For the “post-review to Q1”.
> > >
> > > First, thanks for your acknowledgement that our response clarifies Q1.
> > >
> > > However, we are not sure about the weight sets/instances you mentioned.
> > > (a) If “weight sets/instances” refers to the model (intermediate layers and final output layer) outputs for computing loss and gradients, the two forward passes method indeed needs more memory to save the first forward pass output hidden data in case of being overwritten in the second forward pass. This is what we mentioned about the limitation of computational cost in the paper. (b) If “weight sets/instances” refers to model parameters $w$, we want to point out that the second forward pass will not overwrite $w$ since there is no backward pass (model update) after the first forward pass. The two sub-models used in two forward passes are sampled (dropout) from the same full model, and the same full model parameters are used in the backward pass, only that some parameters are with fewer or zero computations of gradients (see the below example). The second forward pass will not overwrite the first forward pass “weight sets/instances”, and the model parameters of the first forward pass will not need to be stored.
> > >
> > > Take a simple MLP network for illustration, suppose the hidden layer $l$ outputs $h_l = (h_l^1, h_l^2, h_l^3)$ with $3$ neurons, and the connected $l+1$ layer output is $h_{l+1} = h_{l+1}^1$ with $1$ neuron, and the weight parameters are $w=(w_{11}, w_{21}, w_{31})$. The first dropout sample drops the first neuron of $h_l$ to be $h_l=(0, h_l^2, h_l^3)$, and the second dropout sample drops the last neuron of $h_l$ to be $h_l=(h_l^1, h_l^2, 0)$. Then for the first forward pass, the corresponding output of $h_{l+1}$ layer is $h_{l+1}^1=0w_{11}+h_l^2 w_{21}+h_l^3 w_{31}$. Since the model is not updated after the first forward pass (no backward pass and $w$ will not change/overwrite), the $l+1$ layer output for the second forward pass is $h_{l+1}^{1'}=h_l^1 w_{11}+h_l^2 w_{21}+0 w_{31}$. Now for the backward pass, the gradient for weight parameter $\nabla w_{11}=\frac{\partial h_{l+1}^1}{\partial w_{11}} + \frac{\partial h_{l+1}^{1'}}{\partial w_{11}}=h_l^1$ (for $\nabla w_{21}$ and $\nabla w_{31}$, it is the similar calculation). From the equation, we can see that the required stored memories are all the hidden outputs $h_{l+1}^1$, $h_{l+1}^{1'}$, $h_l$ and the original full weight parameters $w=(w_{11}, w_{21}, w_{31})$. There is no need to explicitly store two copies of $w$ for two passes.
> > >
> > > > For the “post-review to Q2 and Q3”.
> > >
> > > We want to clarify that double-batch size will not cause an increase in model parameters $w$. Instead, it will cost nearly 2X of computing and storing model (intermediate layers and output layer) outputs for computing loss and gradients.
> > > For two pass implementation, take NLP task as an example, the input $x$ embedding dimension size is **[batch-size, seq-len, embed-dim]** and the project matrix (weight parameter $w$) dimension size is **[embed-dim, hidden-dim]**. For the double-batch implementation, the input $[x;x]$ embedding dimension size is **[batch-size * 2, seq-len, embed-dim] (not [batch-size, seq-len, embed-dim * 2])** and the project matrix $w$ dimension size is still **[embed-dim, hidden-dim]**.
> > > We speculate that the description of "we repeat the input $x$ once and concatenate them ([$x$; $x$]) in the same mini-batch to forward once." in our paper makes you confused about the concatenated dimension (the batch dimension) and think that "A large batch will require a larger projection matrix (e.g. if $x$ is of size $p$ x $q$ then the larger batch will be $p$ x $2q$)". In your case, $q$ and $2q$ are the batch size, and the projection weight matrix (model parameter $w$) for both are $p$ x **hidden-dim**.  We will make it clearer.
> > >
> > > It can also be seen from training logs (which can be simply checked by running the attached code and the given baseline code). Below we attached the logs.
> > >
> > > ---
> > > **Baseline** Transformer Model (IWSLT14 De->En translation):
> > >
> > > (encoder): TransformerEncoder(
> > >      (dropout\_module): FairseqDropout()
> > >      (embed\_tokens): Embedding(10152, 512, padding\_idx=1)
> > >      (embed\_positions): SinusoidalPositionalEmbedding()
> > >      (layers): ModuleList(
> > >        (0): TransformerEncoderLayer(
> > >          (self\_attn): MultiheadAttention(
> > >            (dropout\_module): FairseqDropout()
> > >            (k\_proj): Linear(in\_features=512, out\_features=512, bias=True)
> > >            (v\_proj): Linear(in\_features=512, out\_features=512, bias=True)
> > >            (q\_proj): Linear(in\_features=512, out\_features=512, bias=True)
> > >            (out\_proj): Linear(in\_features=512, out\_features=512, bias=True)
> > >          )
> > >          (self\_attn\_layer\_norm): LayerNorm((512,), eps=1e-05, elementwise\_affine=True)
> > >          (dropout\_module): FairseqDropout()
> > >          (activation\_dropout\_module): FairseqDropout()
> > >          (fc1): Linear(in\_features=512, out\_features=1024, bias=True)
> > >          (fc2): Linear(in\_features=1024, out\_features=512, bias=True)
> > >          (final\_layer\_norm): LayerNorm((512,), eps=1e-05, elementwise\_affine=True)
> > >        )
> > >
> > >  INFO | fairseq\_cli.train | task: RDropTranslation
> > >  INFO | fairseq\_cli.train | model: TransformerModel
> > >  INFO | fairseq\_cli.train | criterion: RegLabelSmoothedCrossEntropyCriterion
> > >  INFO | fairseq\_cli.train | num. **model params: 36,741,120 (num. trained: 36,741,120)**
> > >
> > > ---
> > >   **R-Drop** Model (IWSLT14 De->En translation):
> > >
> > >   (encoder): TransformerEncoder(
> > >      (dropout\_module): FairseqDropout()
> > >      (embed\_tokens): Embedding(10152, 512, padding\_idx=1)
> > >      (embed\_positions): SinusoidalPositionalEmbedding()
> > >      (layers): ModuleList(
> > >        (0): TransformerEncoderLayer(
> > >          (self\_attn): MultiheadAttention(
> > >           (dropout\_module): FairseqDropout()
> > >            (k\_proj): Linear(in\_features=512, out\_features=512, bias=True)
> > >            (v\_proj): Linear(in\_features=512, out\_features=512, bias=True)
> > >            (q\_proj): Linear(in\_features=512, out\_features=512, bias=True)
> > >            (out\_proj): Linear(in\_features=512, out\_features=512, bias=True)
> > >          )
> > >          (self\_attn\_layer\_norm): LayerNorm((512,), eps=1e-05, elementwise\_affine=True)
> > >          (dropout\_module): FairseqDropout()
> > >          (activation\_dropout\_module): FairseqDropout()
> > >          (fc1): Linear(in\_features=512, out\_features=1024, bias=True)
> > >          (fc2): Linear(in\_features=1024, out\_features=512, bias=True)
> > >          (final\_layer\_norm): LayerNorm((512,), eps=1e-05, elementwise\_affine=True)
> > >        )
> > >
> > >  INFO | fairseq\_cli.train | task: TranslationTask
> > >  INFO | fairseq\_cli.train | model: TransformerModel
> > >  INFO | fairseq\_cli.train | criterion: LabelSmoothedCrossEntropyCriterion
> > >  INFO | fairseq\_cli.train | **num. model params: 36,741,120 (num. trained: 36,741,120)**
> > >
> > > ----
> > >
> > > From the logs, we can see that the double batched training has exactly the same model parameters as baselines, both are 36,741,120 parameters.
> > >
> > > Look forward to your further feedback since this is the end of discussion day.
> > >
> > > We sincerely hope that our understandings are in the same stage about the model parameter/weight.

---

> > > ### Author Response · Authors · 2021-09-03
> > > **Look forward to further response**
> > >
> > > Dear Reviewer a8g8,
> > >
> > >
> > > Thanks for your post-response again.
> > >
> > > Could you check our response to your post-response?
> > > We sincerely hope that our understandings are in the same stage about the model parameter/weight.
> > >
> > > Appreciate it a lot.

---

> ### Author Response · Authors · 2021-08-25
> **Look forward to post-rebuttal feedback**
>
> Dear Reviewer a8g8,
>
> Thanks for your review comments again.
>
> We have posted detailed responses to your questions for two weeks. Could you kindly check the rebuttal? We look forward to your post-rebuttal feedback.
>
> Appreciate it a lot.

---

### Official Review · Reviewer_BNsZ · 2021-07-20

**Rating:** 6
**Confidence:** 4

**Summary:**

This paper proposed to forward an input to neural networks twice with two different dropout masks, and apply a consistent loss as regularization, named R-Drop, between the two outputs to penalize the discrepancy. KL divergence of the two output probabilities is used as the consistent loss. Experiments on different NLP tasks demonstrate the effectiveness of R-Drop.

**Main Review:**

Strengths:

The proposed R-Drop is simple and effective, achieving impressive improvements on several tasks.

Weakness:

The main issue of this paper is that the authors completely ignore the existing research on regularizing the consistency of dropout. The research line can be back to the work of Ma et al., (2017), which analyzed the gap of dropout between training and inference time, and also proposed to regularized dropout neural networks by adding a loss to penalize the gap.
Subsequently, Zolna et al., (2018) proposed Fraternal Dropout, which also penalize the discrepancy of two dropout masks. The only difference between Fraternal Dropout and R-Drop is that Fraternal Dropout used L2 distance between the hidden representations of the final layer as loss while R-Drop used KL-divergence.
Recently, Gao et al., (2021) applied the same framework in contrastive learning by formulating the outputs with two dropout masks as positive pairs. It is the authors' responsibility to comprehensively discuss the relations and differences between R-Drop and the existing methods.

==============================Update after Rebuttal==============================================
Thanks for the authors' efforts on the response. It is highly appreciated that the authors provided detailed discussions of the missed related work and promised to revise the paper. This has addressed my concerns and I upgraded my score to 6.

References:

Ma et al., Dropout with Expectation-Linear Regularization. 2017.

Zolna et al., Fraternal Dropout. 2018.

Gao et al., SimCSE: Simple Contrastive Learning of Sentence Embeddings. 2021.

**Time Spent Reviewing:**

5

---

> ### Author Response · Authors · 2021-08-10
> **Response to Reviewer BNsZ**
>
> Thanks for your review comments and pointing out the related works. We are sorry that we missed some works in the submission. We will definitely add them in the new version and discuss the relations and differences between R-Drop and them. Here, we first give a discussion and comparison as follows.
>
> 1. **Comparison of ELD (Ma et al., 2017) and R-Drop:**
> While both ELD and R-Drop focus on reducing the training-inference gap, there are some key differences.
>     * ELD works on directly reducing the gap between the sub model with dropout (train) and the expected full model without dropout (inference), while R-Drop works between the sub models only (consistency between two sub model). Directly working on sub models and the full model is harder than on sub models, as discussed and proved in FD (Zolna et al., 2018). FD demonstrated that "simultaneously backpropagating target loss through both networks yields both performance gain as well as convergence gain" (in FD paper section 3.1). Therefore, R-Drop is more favorable that performs consistency regularization between sub models.
>     * ELD is less efficient than R-Drop. ELD minimizes the regularization term by only back-propagating the gradients through the sub model with dropout, and the full model without dropout is not updated. In comparison, R-Drop updates the model parameters with consistency regularization on both sub models with different dropout samples (our bidirectional KL-divergence). Besides, ELD minimizes only one sub model with negative log-likelihood $\log P(y_i |x_i, s)$ (with ground-truth label $y_i$ and dropout sample $s$), while R-Drop trains on two sub models with dropout by minimizing negative log-likelihood $\log P(y_i|x_i, s_1) + \log P(y_i|x_i, s_2)$. Hence, R-Drop training is more efficient than ELD.
>     * Another major difference is the loss function for regularization. ELD uses $L2$ distance between the final hidden states of the full model and sub models for regularization. However, this is not a cogitative choice. It is clear that the main training objective is to minimize the negative log-likelihood with ground-truth $-\log P(y_i|x_i)$, which is built on the prediction probability. However, regularizing the hidden states distance is not in the same space as regularizing the probability distribution since **log-softmax** hugely affects the optimization space and the optimization difficulty. The scale of the cross-entropy loss and the $L2$ distance loss is also different, which makes it harder to tune the coefficient of the $L2$ regularization loss. In comparison, R-Drop utilizes the KL-divergence between the output probability distributions as the consistency regularization, which is in the same space as the training objective (likelihood). More details are discussed in the following comparison between R-Drop and FD.
>
> 2. **Comparison of FD (Zolna et al., 2018) and R-Drop:**
> FD is the most related one with R-Drop, and we are both working on penalizing the discrepancy of two dropout samples. Despite this, the difference of consistency loss between FD and R-Drop is critical.
>    * As we discussed above, the $L2$ distance on hidden states is not in the same space as the KL-divergence between the probability distributions. Since the **main training objective** is to minimize the negative log-likelihood on the output distribution, KL-divergence regularization is more consistent, and the loss scale is more similar to the main training objective. This eases the optimization process, and the consistency regularization can be easily controlled.
>     * Besides, though FD and R-drop are both making the sub models with dropout to be more consistent, $L2$ loss has some problems to achieve this goal. The **sub-model consistency** should be controlled on the output probability level since the main objective of classification is to make the prediction distribution to be closer to the ground-truth distribution. The smaller of the hidden states distance cannot guarantee the model outputs to be closer. Here we show an example: $h_1=(2, 3)$ and $h_2 = (3, 4)$ are two outputted hidden states of the sub models, the $L2$ distance is $d(h_1, h_2) = 2$, which is not zero, but the **softmax** probabilities are same for both sub models so that the KL-divergence $D_{KL}(p_1||p_2) = 0$. In this case, the two sub models are indeed making consistent decisions with each other since the probability distributions are the same, which needs no more regularization as $L2$ loss. Furthermore, if $h_1'=(2,3)$ and $h_2'=(3,3.5)$, the $L2$ distance $d(h_1', h_2')=1.25$, and the $D_{KL}(p_1' || p_2')$ is not zero. Compared to the first case, the $L2$ distance is regularized to be reduced, but the KL-divergence $D_{KL}$ becomes larger, which means the prediction distributions from two sub models are more different, not consistent. This is not the regularization effect that we want.
>
> As summary, our consistency regularization loss is more suitable to achieve the goal of consistency regularization. We also provide supportive experimental comparisons in response 4.
>
>
> 3. **Comparison of SimCSE (Gao et al., 2021) and R-Drop:**
> The SimCSE work is related but not close to our work, except that each data sample is passed to the model with two dropout samples. The key differences are as follows.
>     * The motivations behind SimCSE and R-Drop are very different. SimCSE studies the contrastive learning with a more efficient training on similar (positive) samples so that their representations can be much closer in the embedding space, and the dissimilar ones are far apart from others. Therefore, SimCSE views the two dropout samples as data augmentation samples. In comparison, R-Drop studies the dropout method itself, which works on the sub-model level instead of the embedding representations, the goal is to make the probability prediction to be consistent between sub models, indirectly improving the consistency between sub models and the full model.
>     * The loss is also different. SimCSE uses the contrastive loss between positive samples and negative samples as the training objective, while R-Drop uses the consistency loss as a regularization term instead of the main training objective (the goal is still the cross-entropy loss). Due to the contrastive learning scenario of SimCSE, SimCSE requires the `negative' samples in the loss calculation, while R-Drop only works on the input sample itself.
>     * SimCSE considers cosine similarity based distance on the hidden states, similar to FD and ELD that works on the hidden representations, while R-Drop is on the output probability distribution with KL-divergence loss.
>
> We also want to point out that SimCSE and R-Drop are two contemporary works. SimCSE is released on Arxiv on 18 Apr, while we are busy working on our experiments that time (5 different tasks and 18 datasets, and many of them are large datasets) for NeurIPS submission deadline, 28 May, the gap is one month. We will add those discussions in the new version according to your suggestion.
>
> 4. **Besides the above analysis, experiments are also different.**
> FD works on the language modeling with RNN architecture, while we give a more comprehensive verification: we test on $5$ different tasks (both NLP and CV) with strong Transformer model and pre-trained models (BERT, BART, ViT). To give a better understanding of the advantages of R-Drop over hidden space regularization, we also conduct experimental studies. Since FD has shown its advantage over ELD, we mainly present experimental comparisons between FD and R-Drop. The experiments are image classification on the cifar-10 dataset (used in FD work) and IWSLT14 German->English Translation (used in R-Drop work). The results are shown as follows.
>
> **(1) cifar10 image classification**
>
> | model | Unlabeled data | Test Acc |
> | --------- | ----------- | ---- |
> | baseline   |  No   |  77.1 |
> | FD   |  No   |  77.6 |
> | R-Drop   |  No   |  78.13 |
>
>
> **(2) IWSLT14 German->English translation**
>
> | model | BLEU |
> | --------- |  ---- |
> | baseline   | 34.78 |
> | FD   |   35.04 |
> | R-Drop   |   37.25 |
>
> The cifar10 experiments are conducted on the code (https://github.com/akamaster/pytorch_resnet_cifar10), and we add R-Drop regularization as our implementation. (In the image classification task, the R-Drop model hyper-parameter $\alpha$ needs to reduce linearly with the learning rate implemented by torch.optim.lr\_scheduler.MultiStepLR.)
> The IWSLT experiments are on our released code, and we implement the FD same as its released code. From the tables, we can clearly see that R-Drop is superior than FD on both tasks, which can prove the advantages of the KL-divergence consistency regularization.
>
> Again, thanks for your reviews, and we will definitely put these discussions in the new version. Hope the above answers can help address your concerns.
>
> [1] Ma et al., Dropout with Expectation-Linear Regularization. 2017.
>
> [2] Zolna et al., Fraternal Dropout. 2018.
>
> [3] Gao et al., SimCSE: Simple Contrastive Learning of Sentence Embeddings. 2021.

---

> > ### Comment · Reviewer_BNsZ · 2021-08-25
> > **Re: Response to Reviewer BNsZ**
> >
> > Thanks for the authors' efforts on the response. It is highly appreciated that the authors provided detailed discussions of the missed related work and promised to revise the paper. This has addressed my concerns and I upgraded my score to 6.

---

> ### Author Response · Authors · 2021-08-25
> **Look forward to post-rebuttal feedback**
>
> Dear Reviewer BNsZ,
>
> Thanks for your review comments again.
>
> We have posted detailed responses to your questions and concerns for two weeks. Specifically,  we put detailed analysis, examples, and experimental comparisons to show the difference and advantages between the related works and R-Drop.
>
> Could you kindly check the rebuttal? We look forward to your post-rebuttal feedback and discussion.
>
> Appreciate it a lot.

---

### Decision · Program_Chairs · 2021-09-27

**Decision:**

Accept (Poster)

**Comment:**

The paper presents a method for self-supervising models by ensuring that different forward passes through dropout result in the same prediction distributions. Results show the utility of the method. The reviewers raised really interesting points and brought up prior work but the authors were able to place their work within this context, and show their novelty.  Points raised during the discussion will be added to the paper for publication.